# The Effect of Growth Rate during Infancy on the Risk of Developing Obesity in Childhood: A Systematic Literature Review

**DOI:** 10.3390/nu13103449

**Published:** 2021-09-29

**Authors:** Anela Halilagic, George Moschonis

**Affiliations:** Department of Dietetics, Nutrition and Sport, School of Allied Health, Human Services and Sport, La Trobe University, Melbourne, VIC 3086, Australia; anela.halilagic@outlook.com.au

**Keywords:** infant growth, rapid growth, growth velocity, obesity, overweight, body mass index, childhood

## Abstract

The prevalence of childhood obesity has been trending upwards over the last few decades. Recent evidence suggests that infant growth rate has the potential to increase the risk of obesity development during childhood. This systematic literature review aimed to summarise the existing evidence on the relationship between infant growth rate and subsequent childhood obesity. Studies were sought for that assessed the effect of infant growth rate on outcomes of overweight, obesity, BMI, waist circumference or body composition measures among a population group of children aged 2 to 12 years old. Data sources included PubMed, CINAHL, Web of Science and MedLine. Twenty-four studies were identified as eligible and included in this review, out of 2302 publications. The ADA Quality Checklist was used to assess the quality of individual studies. Ten studies received a positive result and 14 studies a neutral result. A narrative synthesis was completed to present study characteristics and results. Several independent positive associations were determined between rapid growth at different stages during infancy and overweight, obesity, BMI, waist circumference and body composition in childhood. Further investigation is required to determine if a specific period of infancy carries greater associations of risk with childhood outcomes. Determining an ideal rate of infants’ growth as a means to minimise the future risk of childhood obesity should be the focus of future research that will also inform early life obesity prevention strategies. Registration no.: CRD42021244029.

## 1. Introduction

The global prevalence of obesity has been trending upwards over the last few decades, with the current rate reaching almost triple the rate in 1975. Childhood and adolescent obesity specifically, has seen a greater rate increase with the prevalence growing from 4% in 1975 to 18% in 2016 [1]. Childhood overweight and obesity has been found to track or remain stable from infancy through to childhood and adulthood [2]. This rapid increase in prevalence of obesity among children has led to the classification of childhood obesity as an epidemic [3]. 

### 1.1. Health Consequences and Risk Factors of Childhood Obesity

Childhood obesity is associated with higher risks of premature mortality and morbidity in adulthood [4]. It also has many other detrimental health risks including breathing difficulties, hypertension, increased risk of fractures, early markers of cardiovascular disease, insulin resistance and psychological effects. In addition to this, some other health consequences that are often not evident until adulthood are cardiovascular diseases, diabetes, musculoskeletal disorders and certain types of cancers [1]. 

Risk factors are important to recognise in order to understand the underlying causal factors behind childhood obesity. Prenatal and postnatal periods are critical times in which the foetus and infant are exposed to various risk factors that may have potential effects on growth, development and future health status. This period of time from conception to the end of the second year of an infant’s life is known as the ‘First 1000 days’, a period of maximum developmental plasticity [5]. Risk factors that occur within the prenatal period can include overweight or obesity of the mother pre-pregnancy, excessive gestational weight, alcohol consumption and smoking during pregnancy, gestational diabetes and maternal stress. Once the mother has given birth, post-natal risk factors can include a high birth weight, absence of breastfeeding, early or late introduction to solid foods and rapid growth rate during infancy [5]. 

### 1.2. Infant Growth Rate

Infant growth rate can be determined through measurements in terms of weight-for-age, length-for-age, weight-for-length, BMI-for-age percentiles and z-scores, and the velocity reflected to the changes observed in these growth measurements during specific time periods throughout infancy. Most commonly, growth velocity is defined as a change in weight or height over a certain time period, expressed as g/month or cm/month. The expected pattern of infant growth under adequate conditions begins with a rapidly decelerating growth rate from birth, which then reaches a near-plateau by the end of the first year of life and then continues to taper off throughout the second year [5]. For infants born full-term, the rate of infant growth can be classified as retarded growth, normal growth or rapid growth depending on whether growth velocity during infancy shows a steep downward, a stable or a steep upward move of the infant’s position, respectively, in the growth charts. Pre-term infants or infants born small for gestational age (SGA) experience a different pattern of growth, most commonly showing an accelerated growth velocity, known as “catch-up growth”.

The common definition of rapid growth during infancy is a change in weight or length-for-age standard deviation score greater than +0.67 from birth to age 24 months. In contrast, retarded growth is commonly defined as a change in weight or length-for-age standard deviation score greater than −0.67 from birth to age 24 months [6]. Although these definitions are often used, some researchers that have explored rapid and retarded infant growth have used alternative terms and measures as criteria for defining growth.

### 1.3. Current Evidence

Other published systematic literature reviews that were identified assessed the association between infant growth and subsequent obesity risk with varying focuses. Cho et al. [7] and Hong et al. [8] focused their reviews on SGA infants as their population groups of interest. Andrea et al. [9] focused their review on studies that had racial/ethnic minority or low-SES study populations. Rallis et al. [10] focused on studies with only body fat measures as their outcomes. Arisaka et al. [11] explored the effects of rapid weight gain in infancy and childhood, rather than focusing solely on infancy. Finally, Rolland-Cachera et al. [12] combined infant growth and nutrient intake as their exposure when investigating the risks of developing childhood obesity.

The current available evidence highlights the association between infant growth rate and subsequent obesity risk, however there is a limited comprehensive understanding of this relationship that also considers the factors of varying birth weight status, duration of exposure and the relationship between obesity risk factors and the development of obesity.

### 1.4. Objectives

We aimed to identify individual studies that could be analysed and summarised to allow readers to easily access and understand the available evidence surrounding this topic. Therefore, we conducted a systematic literature review to identify and evaluate the association between infant growth rate and subsequent obesity risk in childhood by summarising the findings of relevant studies. We also summarised study findings about whether birth weight and infant age altered the risks associated with the rate of infant growth on future obesity and associated risk factors, such as waist circumference and body fat percentage.

## 2. Materials and Methods

The protocol of this review was submitted for registration and published by PROSPERO on 19 April 2021, titled “The effect of growth rate during infancy on the risk of developing obesity in childhood: a systematic literature review” (registration number: CRD42021244029). The registered protocol can be accessed via: https://www.crd.york.ac.uk/prospero/ (accessed on 19 April 2021). This systematic review was drafted according to the guidelines of the PRISMA statements.

### 2.1. Eligibility Criteria

The studies that were sought for review described a relationship between the rate of infant growth and subsequent risk of obesity development during childhood. The population group of interest were children aged between 2 and 12 years old, inclusively, who had available exposure data on growth rate during infancy. The primary outcomes were overweight, obesity and BMI, with secondary outcomes including anthropometric measurements, such as waist circumference and body composition markers. Studies were eligible for inclusion if they were a peer-reviewed, cohort study in the English language, included human participants and published from the year 2010 onwards. It was important to place a limit on year coverage, 2010 to 2021, to ensure that the studies included were focused on recent growth trends and population groups. Cohort studies selected as the study design for inclusion due to the higher level of evidence they represent and their capacity to prove causal associations, while randomised control trial studies were deemed not suitable in regard to answering the research question. Reviews on similar topics that were deemed relevant had their reference lists screened for eligible studies that may have been missed during the initial searches and were therefore not included in this review. 

The minimum duration of follow up was not set, as the years of infancy directly preceded childhood and therefore, relevant studies may have been excluded if a minimum period was determined. Exclusion criteria deemed studies ineligible if the subject groups were solely preterm infants as this population group experiences a pattern of infant growth that is unlike, and therefore not comparable to, the growth of full-term infants. Studies that involved sample populations that were a mixture of preterm and full-term infants were included due to the value of the results to this review, with data extracted only for full-term infants wherever feasible. Additionally, studies were excluded if the outcomes of interest were not measured or reported, as this would hinder any described associations and provide a lack of basis for comparison against other studies.

### 2.2. Searches & Information Sources

The search for eligible studies was conducted using the PubMed, CINAHL, Web of Science and MedLine databases with identical and appropriate date, language and age limits applied. The Cochrane Library database was also initially searched to identify similar reviews to this current one, with the results identifying none. The last date of searches conducted of each database was 30 March 2021. The key words and medical subject headings (MeSH) used as part of the search strategy are identified in the example in Table 1 below. Terms were combined using the Boolean Operator “OR” for terms within the population, exposure and outcome categories, and “AND” for combining the population, exposure and outcome groups. The searches were limited to “Title and Abstract” to further focus the results to only include studies of most relevance. 

### 2.3. Study Selection

The screening process was in accordance to the published review protocol to ensure that replication could be possible. The database searches were conducted by one reviewer, A.H., which was followed by results being exported to the software, EndNote 20. This software was then used to store the references and identify duplicates to be removed. The remainder of studies, following the removal of duplicates, were then uploaded to Covidence for the initial title and abstract screening process, also performed by author A.H. After excluding further studies based on non-eligibility, the next step of reviewing the full texts was completed by both reviewers, A.H. and G.M. independently, using the inclusion and exclusion criteria to determine the final number of studies to be included in the review. All disagreements were resolved through discussion and agreements between both reviewers. 

### 2.4. Data Collection and Data Items

The data extraction process began with both reviewers, A.H. and G.M., independently completing the first five studies for the purpose of comparison. The extraction process was then continued with one reviewer, A.H., independently completing the remaining number of studies. The comparison of extracted data from the first five studies allowed for any indifferences to be discussed and resolved between reviewers, to then allow for the reviewer A.H. to complete the extraction process in accordance with both reviewers’ combined expectations.

All data was organised in an Excel spreadsheet that was set with a template outlining select variables including the title, journal, year of publication, author names and affiliations, funding sources, conflict of interest, study design, location and setting, aim, inclusion and exclusion criteria, population, exposure, duration of exposure, sample size, recruitment process, retention rate, demographic data, primary and secondary outcomes, calculations and effects on outcomes of the study. These variables were chosen as they provide an adequate summary of the full texts and would allow for ease of information identification for upcoming critical appraisal and data synthesis processes. 

Funding sources were defined as any monetary benefits received by authors from supporting institutions or companies in relation to the completion of the study. This could also be related to author affiliations, which may be any institute that an author belongs to that could also have an interest in the research findings. Another variable extracted from this review’s studies was declaration of conflict of interest. This allows readers to identify potential sources of bias based on funding and affiliations. Missing or unclear data on conflict of interest statements in the studies led to the assumption that the authors were not transparent about potential conflicts, thereby increasing the risk of bias.

Inclusion and exclusion criteria that were not identified in the full text of a study were searched for in mentioned associated studies, such as those that might have been the original studies that samples were recruited from. Demographic data included countries that the research was conducted in and any additional information such as cities, towns, schools and classification of rural or urban populations, where relevant.

The population group was defined as the childhood age range that outcome measurements were taken within, while the exposure duration was defined as the age or range that infant growth measurements were taken at. The sought outcomes were divided into primary and secondary, dependent on whether they were directly related to overweight or obesity, or measures related to risk factors, such as body fat mass or waist circumference, respectively. All results that were identified to be compatible with the primary and secondary outcomes were selected to be included in the synthesis of data. 

### 2.5. Risk of Bias

Risk of bias within individual studies was assessed using the ADA Quality Assessment Tool to evaluate the identification and addressment of potential issues. Each study was assessed against the criteria at an outcome or study level by one reviewer, A.H, and given a positive, neutral or negative result. The areas assessed in the ADA Quality Assessment Tool include the research question, selection and comparability of study groups, withdrawals, blinding, exposures, outcomes, statistical analyses and conflicts of interest. A positive result was assigned if most of the answers to the tool questions were “Yes”, indicating that the study addressed relevant potential bias issues. In contrast, a negative result was assigned if most (six or more) of the answers to the tool questions were “No”, indicating that the issues were not adequately addressed. The resulting positive, neutral or negative appraisal outcome for each report was used to determine the risk of bias affecting the accuracy, relevancy and generalisability of the results and findings.

Potential sources of bias that may have had an effect on the cumulative evidence in this review include publication bias and confirmation bias, whereby the available evidence may present skewed findings based on a lack of publication of unfavourable results. The extend of our search to original studies that provided additional information about the selected study that was not reported in the article identified as part of the review process, were also used as basis for the identification of risk of bias across studies and specifically for publication and selective reporting bias.

### 2.6. Data Synthesis

The main summary measures that were assessed and synthesised in the results relating to categorical outcome measures (i.e., obesity and overweight) were odds ratios (OR) and incidence rate ratios (IRR). Beta coefficients, means and correlation coefficients were the summary measures identified for the continuous outcome measures (i.e., BMI, waist circumference, body fat percentage, fat mass index and fat free max index). These measures allowed for the effects of exposures on outcomes to be identified and the results to be easily interpreted for discussion. 

Extracted study data was narratively synthesised into a table highlighting the key features of each study. The synthesised data included information on the study design, location, sample size, infant growth measure, age range of exposure, childhood growth measure and age range of measurements, analysis, size of effect for primary and secondary outcomes, and risk of bias assessment. The summary measures are located in the size of effect columns to allow for comparison between studies. The order of the studies presented in the summarised table was organised based on outcomes, with studies that reported on overweight or obesity first, followed by BMI, waist circumference and body composition.

Narrative synthesis allowed for a summary to be developed that was used to determine the extent, quality and applicability of current evidence available on the research question of interest. This involved the identification of similarities and differences between studies, the assessment of the effects of exposures on outcomes and the strength of quality of each study. A meta-analysis was not appropriate nor feasible at the time that this review was written, however it could be a future option to further improve the strength of quality. As this review focused on a narrative method to synthesise results, the use of measures of consistency to determine heterogeneity between studies was not required.

## 3. Results

### 3.1. Study Selection

We identified 3060 references from the initial search of databases. Scanning for duplicates using EndNote 20 resulted in the identification and exclusion of 758 duplicate references. The remaining 2302 references were involved in the title and abstract screening process, which identified 2236 irrelevant studies to be excluded due to a wrong population, exposure, outcome or study design. The number of references that underwent a full text review, based on inclusion and exclusion criteria, was 66. Following this, 42 studies were further excluded due to not meeting the criteria, with exact reasons outlined in the flow diagram (Figure 1). The final number of studies to be included in the review was 24, comprising of 15 prospective cohort studies [13,14,15,16,17,18,19,20,21,22,23,24,25,26,27], 6 retrospective cohort studies [28,29,30,31,32,33], and 3 longitudinal cohort studies [34,35,36]. These 24 studies have been presented in Table 2.

### 3.2. Study Characteristics

The locations of the studies ranged across six continents, including one in Australia, six in Asia, two in Africa, seven in Europe, five in North America and three in South America. In addition to this, the countries varied in economic developmental status, including developed countries such as The Netherlands, and developing countries such as Malawi.

Two studies presented results that were not statistically significant, however their methodological data and study characteristics have been included in the following analysis. The population groups among all the studies ranged from 2 to 12 years of age, including both boys and girls. Thirteen studies had population groups that had an age range with a minimum one year and maximum six-year range [13,17,23,24,25,26,28,29,30,32,33,34,36]. The remainder of the studies presented their sample populations with one age, such as an example of seven years old. When combining all the ages studied, approximately 60% was within the range of 6 and 10 years. Only three studies had population groups that included 2 years of age [25,26,32], while only another four studies included the ages of 11 and 12 years [13,19,24,34].

The examined outcomes that produced statistically significant results among the studies included a range of primary and secondary, as well as singular and multiple outcomes grouped together. The outcomes included overweight and obesity, BMI, waist circumference and body composition. Within these outcomes there were subcategories and measures. Fourteen studies explored the primary outcomes [13,18,19,20,22,23,26,29,30,31,32,34,35,36], three studies explored the secondary outcomes [14,25,27], and seven studies explored both the primary and secondary outcomes [15,16,17,21,24,28,33]. Outcome measures included odds ratios, incidence rate ratios, beta coefficients, means and correlation coefficients. Statistical significance was determined based on presented *p*-values (i.e., when *p* < 0.05) or 95% confidence intervals.

### 3.3. Main Exposures

Growth rate during infancy was determined by measures of change in weight and length over a given duration between birth and 2 years of age. All studies identified exposure durations that began at birth or 0 months. The minimum duration of exposure was 6 months, and the maximum duration of exposure was 24 months. Only one study’s exposure was 0–6 months [16], with one other study having an exposure of 0–9 months [14]. Other durations included 0–12 months found in 11 studies [13,15,19,20,21,22,23,25,27,34,35], 0–18 months in 3 studies [17,28,29] and 0–24 months in 8 studies [18,24,26,30,31,32,33,36]. From these values, it can be summarised that approximately half of the studies measured the determined exposure in the first year of life, and the other half of the studies measured the exposure in the first two years of life.

The studies in this review measured their exposures in many ways, including weight gain, length gain, BMI gain, weight velocity, height velocity, BMI velocity, and changes in weight-for-age z-score (WAZ), height-for-age z-score (HAZ), weight-for-height z-score (WHZ) and BMI z-score (BMIZ). Weight velocity and height velocity were measured in terms of kg/month and cm/month, respectively. Changes in standard deviation score (z-score) were measured by determining the difference in z-score between two data points. This difference was then compared against growth rate definitions to determine the rate of growth of an infant. 

### 3.4. Variations of Definitions

Among the studies investigating overweight and obesity as outcomes, the definitions varied and therefore had an impact on the comparability of results and findings. Studies by Zhou et al. [17], Penny et al. [35] and Shi et al. [32] defined overweight as a BMI-for-age z-score (BAZ) > +1 but < +2 and obesity as BAZ ≥ +2. Nguyen et al. [23] defined a BMIZ > 1 as overweight/obesity. Similarly, Jones-Smith et al. [18] focused on overweight, and not obesity, as an outcome and defined overweight as BAZ > +1. Other studies defined overweight and obesity based on BMI percentiles, with Lei et al. [20] defining a BMI > 85th percentile but <95th percentile as overweight and BMI > 95th percentile as obesity. Likewise, Taveras et al. [36] defined obesity as BMI ≥ 95th percentile, while Woo et al. [33] defined a BMI ≥ 85th percentile as overweight/obesity. Other studies such as those by Nanri et al. [29] and Taal et al. [26] defined overweight and obesity according to the suggested BMI cut-off points proposed by the International Obesity Task Force (IOTF). Variations in definitions were due to the use of a range of international and national growth charts (i.e., WHO, IOTF, CDC and other country-specific) to assess childhood growth, overweight and obesity.

Definitions of rapid weight gain were also not identical among all studies as there were differences in the identified time frame that the measured growth occurs in. Zhou et al. [17] and Nanri et al. [28] defined rapid weight gain as a change in WAZ > +0.67 between birth and 1.5 years. Other studies by Lin et al. [21] and Penny et al. [35] did not identify a time frame that growth is required to occur in for the growth rate to be defined as rapid weight gain. Rather, they defined rapid weight gain as a change in WAZ > 0.67 over two time points. Gishti et al. [15] defined an SDS change greater than 0.67 between different time points as growth acceleration, also familiarly classified as rapid weight gain.

Catch-up growth was a term used by Kramer et al. [34] to define an increase in WAZ > 0.67, the same definition frequently used for rapid weight gain. Similarly, Taal et al. [26] defined catch-up and catch down growth as a change in SDS of >0.67 from birth to 2 years of age. Shi et al. [32] defined rapid catch-up growth as WAZ ≥ 0 but ≤1, and excessive rapid growth as WAZ ≥ 1 within the first two years of life. 

### 3.5. Status of Birth Weights

A select number of studies identified the birth weight status of their sample population as either small for gestational age (SGA), appropriate for gestational age (AGA), large for gestational age (LGA) or a combination of the three classifications. Similar to obesity and overweight, the definitions of SGA, AGA and LGA were not entirely alike among all the applicable studies. Kramer et al. [34] defined a birth weight < 10th percentile as SGA and a birth weight > 90th percentile as LGA. Lei et al. [20] also defined SGA as a birth weight < 10th percentile. In contrast, Taal et al. [26] and Vogelezang et al. [27] defined SGA as birth weight < 5th percentile and LGA as a birth weight > 95th percentile. Other studies that did not provide information about the birth weight status of their study samples were assumed to have not identified this during their data collection process, consequently leading to the assumption that samples were a combination of children born SGA, AGA and LGA.

### 3.6. Studies on Childhood Overweight and Obesity

Thirteen studies assessed the relationship between the rate of infant growth and subsequent childhood overweight and obesity. Four of the thirteen studies focused on overweight individually as the outcome [13,18,29,30] and another two studies focused on obesity as the sole outcome [22,36]. The remaining seven studies had childhood overweight and obesity combined as their outcomes [17,20,23,26,32,33,35]. The varying measured exposures associated with overweight and obesity included weight velocity in three studies [13,18,33], rapid weight gain or catch-up growth in seven studies [17,20,26,29,32,35,36], and the remaining three studies measuring weight gain and BMI trajectory [22,23,30]. Duration of exposure varied from 6 to 24 months, with six studies using 0–24 months [18,26,30,32,33,36] and five studies using 0–12 month periods [13,20,22,23,35]. The ages of sample populations ranged from 2 to 11 years old, with 75% of the studies having a sample that was between 6 and 11 years old.

Twelve of the thirteen studies presented results that demonstrated statistically significant, positive associations between the infant growth parameters and outcomes, finding that infants who grew at a faster rate during the first two years of life were at greater risk of developing overweight and obesity in childhood. Chirwa et al. [13] analysed the effect of their exposures by splitting the duration into 0–3 months, 0–6 months and 0–12 months. OR (95% CI) showed positive associations but values decreased as the duration increased; 0–3 months: 4.80 (2.49, 9.26), 0–6 m: 2.60 (1.77, 3.83), 0–12 m: 2.46 (1.89, 3.61). Likewise, Nguyen et al. [23] found a similar trend when interpreting their results that showed that the IRR (95% CI) value was also showing positive associations but was decreasing as age of exposure increased; 0–<6 m: 1.35 (1.06, 1.72), 6–<12 m: 1.40 (1.09, 1.81). Nanri et al. [29] also presented results that identified a positive association between infant rapid weight gain and subsequent obesity at ages 9 to 10, however these findings were not statistically significant.

Of the studies that presented statistically significant results, all except one study presented odds ratio values that reflected positive associations, with values ranging from 1.03 (95% CI: 1.01, 1.05) to 11.6 (95% CI: 8.8, 15.3). Nguyen et al. [23] presented their significant positive association between weight gain and overweight/obesity as an incidence rate ratio value of 1.35 (95%: 1.06, 1.72). 

### 3.7. Studies of Childhood BMI

Eleven studies explored the association between infant growth rate and childhood BMI. Four studies [17,19,26,31] and six studies [15,16,21,24,28,34] specified BMI z-score and BMI as their outcome measure, respectively. One study presented statistically significant results for both BMI and BMIZ outcomes [13]. In contrast, Kramer et al. [34] presented results that were not statistically significant. Rapid infant growth was the exposure measure for four studies [16,17,21,28], while the remaining study measures were weight velocity, height velocity, change in WAZ, change in WLZ, accelerated and decelerated growth, and catch-up growth.

All of the studies presented results that demonstrated positive associations between the individually selected exposure growth measures and childhood BMI, except for one that identified a negative association among infants born SGA [34]. Additionally, one of these studies [15] also presented a negative association between decelerated growth and childhood BMI, with a result of B:−0.35 (95% CI: −0.45, −0.26). Nine studies presented beta coefficient values to represent the strength of the effect of infant growth on childhood BMI, with a range from 0.19 (95% CI: 0.03, 0.35) to 17.4 (SE: 1.37). In addition to other results that mainly analysed the association between weight change and subsequent BMI, Chirwa et al. [13] also presented results on the effect of infant height velocity on childhood BMIZ. The interpretation of results finds that there was a positive association between height velocity and BMIZ (B: 1.50 (SE: 0.54)).

In contrast to the results previously mentioned for the overweight/obesity outcome, Chirwa et al. [13] presented results of beta coefficient (SE) values for split exposure durations, 0–3 months, 0–6 months and 0–12 months. The results show that as the duration during infancy for which growth rate has been assessed increases, from 3 months to 12 months, the effect of infant weight velocity on childhood BMI also increases, as shown by the change in B(SE) values from 12.3 (1.05) to 17.4 (1.37), respectively. Another study’s results that present findings unlike others in the review is by Polk et al. [31], that showed that there was a greater positive association between infant change in WLZ and childhood BMIZ among boys than girls, with B (95% CI) values of 1.77 (1.34, 2.23) for boys and 1.03 (0.55, 1.51) for girls. A multivariate linear regression was used to factor in gender as a variable to examine the relationship of change in WLZ and childhood BMIZ among boys and girls.

### 3.8. Studies on Waist Circumference

Waist circumference was assessed as an outcome in three studies that explored the effect of rapid weight gain during infancy on childhood waist circumference. The three studies had three different exposure durations: 0–6 months [16], 0–12 months [21] and 0–18 months [28]. Additionally, the ages of the sample populations at outcome were 8 years, 4 years and 9–10 years, respectively. Exposures included infant weight gain >0.67 SDS [16] and rapid infant growth [21,28]. All identified results were statistically significant, with two studies presenting results as B (95% CI) values [16,21], 4.0 (2.1, 5.9) and 1.62 (0.45, 1.32), and the remaining study results as adjusted means (95% CI).

### 3.9. Studies of Body Composition

Nine studies investigated the effect of infant growth rate on body composition markers during childhood. Markers that were identified included fat mass index (FMI) in five studies [14,15,17,25,33], body fat percentage (%BF) in three studies [24,28,34], FFMI in one study [25], and visceral fat in another study [27]. Growth exposures included rapid weight gain, weight velocity, change in WAZ and catch-up growth. Beta coefficients, adjusted means and correlation coefficients were used as outcome measures. The range of beta coefficient values was from 0.21 (SE: 0.05) to 0.58 (95% CI: (0.37, 0.80)). Identified results presented in eight of the studies showed statistically significant, positive associations between infant growth rate and body composition markers. Kramer et al. identified a negative association between infant catch-up growth and %BF among infants born SGA, however these results were not statistically significant. One study also measured the effect of decelerated infant growth on FMI during childhood and found an association (B: −0.22 (−0.32, −0.13)) that showed the opposite effect of accelerated growth [15]. Two studies had split exposure durations and presented results on the effect of change in infant WAZ on FMI (0–5 m: B: 0.21 (SE: 0.05), 5–9 m: B: 0.42 (SE: 0.06), *p* < 0.001, [14]) and %BF (0–3 m: B: 0.38 (SE: 0.69), *p*: 0.048, 3–24 m: B: 1.71 (SE: 0.64), *p*: 0.008, [24]). 

### 3.10. Risk of Bias

Following assessment of the risk of potential bias in each study based on the ADA Quality Checklist, 10 out of the 24 selected studies received a positive result, indicating a low risk of bias, while 14 studies received a neutral result, indicating a medium risk of bias. No studies were assessed as having a high risk of bias. The main sources of bias identified among many studies were high withdrawal rates, incomplete data sets, conflicts of interest and poor generalisability. Withdrawal rates were calculated by dividing the number of participants included in the final analysis by the number of participants initially recruited in a study. A withdrawal rate > 20% was assessed as a risk for potential bias. Six studies identified that their data sets were incomplete, mainly due to participant withdrawals throughout the study [13,15,22,26,27,36]. Potential conflicts of interest were identified among studies that did not provide a statement pertaining to no declaration of conflict, or where statements were not clear enough to determine that there was no conflict of interest. Poor generalisability of results was also determined based on statements found in studies regarding acknowledgement of the reduced generalisability of their findings, with the main causes including small sample size, convenient sampling, and inadequate population representation. The quality assessment results and sources of bias for individual studies are presented in the synthesised table. There was no identified risk of bias across studies, specifically relating to publication and selective reporting bias.

## 4. Discussion

The previously highlighted study characteristics identified the age range of the populations as well spread out across early childhood to preadolescence. This supports the accuracy of the initial search strategies to enable results to be limited to studies most relevant to the population of interest defined in the PECO (i.e., Population, Exposures, Comparator, Outcome) framework. It also allowed for effects on outcomes to be identified and compared across the childhood ages of 2 to 12 years, to outline possible trends of size of effect. The numerous countries that the studies were based in allowed for the interpretation that the effect of rate of infant growth is not limited to certain population groups, but is rather applicable globally, similarly to the prevalence of the obesity. Due to the differences in exposure durations and related infant ages found among the studies, the opportunity to explore the effects of these differences on outcome measures became feasible. 

Approximately 60% of the studies included in the analysis of this review were prospective cohort studies, which have the potential to provide the strongest level of evidence compared to other observational studies. This majority provides strength and quality to overall body of evidence in this review. The quality of the included studies has the potential to impact the identified limitations and implications, therefore improving the final quality of this paper. 

The primary and secondary outcomes defined in the PECO framework were identified individually or as a combination within all studies. The availability of multiple pieces of evidence and sets of results for each outcome allowed for the ability to compare findings and improvement of the quality of interpretation and future indications.

### 4.1. Overweight and Obesity

The evidence presents results that demonstrate a positive relationship between infant growth rate and overweight and obesity during childhood among a range of ages and exposure durations. Studies found that infants that experienced rapid weight gain, catch-up growth or increased weight velocity were all associated with a subsequent increased risk of developing obesity or being overweight during childhood. There were no identifiable trends of change in risk among the research of different durations of exposure on overweight/obesity. In regard to age at outcome, two studies found that children at a younger age, i.e., at 3 and 5 years, who experienced rapid infant growth, had greater odds of developing obesity than during later childhood, i.e., at 7 and 10 years, respectively [33,36]. When analysing results that presented split exposure durations [13], it was found that the risk of overweight was greater associated with weight velocity within the first 3 months than within the first 12 months of infancy. This finding suggests that there are greater risks associated with infant growth rate in the earliest period of infancy than later on. The consistency of associations among the studies relating to overweight and obesity were also seen in a range of settings in developed and developing countries. It was found that infants who experienced rapid infant weight gain were at greater risk of developing overweight or obesity if they were in a rural setting, compared to an urban setting [35]. Although this was a significant finding, only one study explored this exposure and outcome relationship among an urban and rural setting.

### 4.2. Body Mass Index

Results suggest that infant growth rate is positively associated with childhood BMI, suggesting that rapid growth during infancy increases the risk of a high BMI during childhood. Growth of varying durations, between birth to 2 years of age, that were measured as weight velocity, weight change, catch-up growth or rapid infant weight gain were related to BMI outcomes up to 12 years of age. Two studies analysed their results based on birth weight status, however only one of these presented statistically significant results [26]. This study found that like other evidence in this review, there was a positive association between catch-up growth and BMI among AGA infants, however, infants that were born SGA experienced a negative association [26]. Likewise, the second study also identified a negative association between catch-up growth and BMI among infants born SGA [34]. Although not statistically significant, there may be some clinical significance in this supporting finding. A decelerated rate of growth and catch-up growth in SGA infants were independently found to be associated with a decreased risk of developing a high BMI [15,26]. 

### 4.3. Waist Circumference

Results on the relationship between infant growth rate and childhood waist circumference support the finding that there is a positive association between the exposure and outcome. The results suggest that an increased rate of infant growth increases the risk of developing a high waist circumference during childhood. As the three studies that explored this association had varying exposure durations, 0–6 months [16], 0–12 months [21] and 0–18 months [28], it can be stated that rapid infant growth of any duration during the first 18 months of life has a positive association with childhood waist circumference. Additionally, the sample group ages ranged from 4 to 10 years, suggesting that the effect of rapid growth during infancy is not restricted to a specific age during childhood, but can rather be observed throughout childhood. 

### 4.4. Body Composition

Childhood body composition measures, such as FMI, FFMI, %BF and visceral fat index, were mostly found to be positively associated with infant growth rate. The results indicate that a greater rate of infant growth is likely to lead to greater body fat mass levels in childhood. Duration of exposure varied among the studies, however this was not found to have an effect on the outcomes. One study identified a negative association between infant growth velocity at 6–12 months and FFMI z-score at 2–3 years of age, in contrast to a positive association between the same outcome and infant growth velocity at 0–6 months [25]. These findings raise the question of whether infant growth in the earliest stage of infancy may have a greater effect on childhood body composition outcomes. 

Additionally, and similarly to the findings on WC, the age range that outcomes were explored within throughout childhood were not limited to early, middle or late childhood. This highlights that the effects of rapid infant growth can be observed across the years of 2 to 12. In regard to decelerated growth, results from one study identified that decelerated infant growth was negatively associated with FMI at 6 years of age [15]. This result was the only one within the studies in this review that presented a statistically significant, negative association between retarded or decelerated growth and the outcome of interest. This finding highlights a rate of infant growth, other than rapid, that could also potentially have a causal relationship with adiposity measures in childhood.

### 4.5. Primary and Secondary Outcomes

All studies that explored the effect of infant growth rate on primary and secondary outcomes presented results that indicated a correlation between the two outcome groups. Positive associations found among primary outcome results were also found among secondary outcome results. For example, rapid infant growth rate that was found to be positively associated with the primary outcomes of BMI and overweight/obesity, was also found to be positively associated with any secondary outcomes investigated, such as FMI [17]. In contrast, one study that explored the effects of decelerated infant growth identified significant negative associations with BMI and FMI [15]. This relationship between primary and secondary outcomes suggests that the effects of infant growth rate are not limited to affecting one outcome, but rather can affect a range of outcomes and risk factors related to obesity. 

### 4.6. Effect of Birth Weight Status

Only one study presented statistically significant results that allowed for the comparison of association between exposure and outcome among SGA and AGA infants. Although this study also included LGA infants in their sample, the results for this sub-sample were not statistically significant. The results identified a negative association between infant catch-up growth and child BMIZ among infants born SGA, while there was a positive association between the same exposure and outcome among infants born AGA. The normal pattern of growth for SGA infants is to experience catch-up growth, however this is not the normal pattern for AGA infants. Due to this, AGA infants that experienced rapid growth were at greater risk of developing a higher BMIZ score in childhood than SGA infants who experienced catch-up growth. Other studies that identified the birth weight status among their sample population either did not perform any stratified analysis based on the size at birth or did not present significant results, and therefore no comparison could be made.

### 4.7. Exposure Duration

Each of the exposures explored in this review varied in duration, ranging between 6 and 24 months. This was also evident when studies were categorised based on outcome, where multiple exposure durations were identified among the groups. For the majority of results included in the analysis, it was shown that outcomes, such as overweight/obesity and BMI, were positively associated with the exposures, regardless of duration. 

Other studies that split their exposure durations into intervals, such as 0–<6 months and 6–12 months, were able to identify differences among the intervals in sizes of effect on their outcomes. A number of studies highlighted that size of effect was greater in the earliest months of infancy, compared to later months, while other study results suggested that size of effect was greater in later months of infancy. Due to the analysis in the review being limited to a narrative analysis, other trends among durations of exposures and outcomes were not observed. 

### 4.8. Limitations

This review used standard methods recommended for systematic literature reviews in accordance to the PRISMA statement [37], however there were some challenges in interpreting the evidence found. Each study had some level of degree of potential risk of bias, which affected the interpretation of results and reduced the applicability for some findings. Exposure and outcome definitions varied among studies causing some difficulty in completing direct comparisons of results. Some studies identified the birth weight statuses of their samples, while others did not, again causing some difficulty in directly comparing results. Two studies included a mixed sample of preterm and full-term infants, which likely had an effect on their results due to the difference in growth patterns. Another limiting factor was that energy intake of the mother and baby during the prenatal and postnatal periods was not investigated in this review, even though it is an important obesity risk factor. However, as growth during infancy is inherently reflective of energy intake at these early life stages, energy intake was indirectly taken into consideration in the current review, since growth was the main exposure examined in the selected studies. The final limitation to mention is the different exposure durations and intervals among studies. 

Although systematic reviews have the ability to provide evidence of very high quality, they are also prone to limitations and bias. One potential source of bias in this review is publication bias, which may have been reduced if any unpublished studies were identified. Although we attempted to reduce any risk of bias, no unpublished studies were identified nor selected for this review. 

### 4.9. Implications and Recommendations

Our findings align with those of previous systematic reviews on similar topics that found positive associations between infant growth rate and subsequent risk of childhood overweight and obesity. Additionally, positive associations were found among other outcomes, including BMI, WC and body composition measures. This understanding of the risks associated with the rate of infant growth is critical for informing recommendations for future research and public health policy. The effects of growth rate during different stages of infancy should be further explored to better understand if a certain period of infancy, such as the first three months, is more critical and associated with greater risks. The effects of infant growth rate on early and late childhood should also be investigated further, as this was an area found to be limited in this review. Other prenatal and postnatal factors could potentially also be included in future analyses, such as maternal gestational weight gain and feeding patterns (i.e., exclusive breastfeeding versus exclusive feeding with formula versus mixed feeding with breastmilk and formula) of infants, to identify possible associations. Finally, retarded or decelerated infant growth rate was another area in this review with limited research, which should be further explored to identify the effect on the risks associated with childhood obesity and risk factors. Determining an ideal rate of growth to minimise the risks of developing childhood obesity should be the focus of future research.

## 5. Conclusions

The associated risks of infant growth rate were clearly identified among the studies in this review. Although the findings demonstrated a clear trend of an increase in childhood obesity risk when infants were exposed to rapid infant growth, further research should be completed to identify additional trends. With limited evidence provided on certain areas, such as duration of growth exposure, changes in risk based on childhood age, and retarded growth, public health policy should be advised and supported by additional high-quality research.

## Figures and Tables

**Figure 1 nutrients-13-03449-f001:**
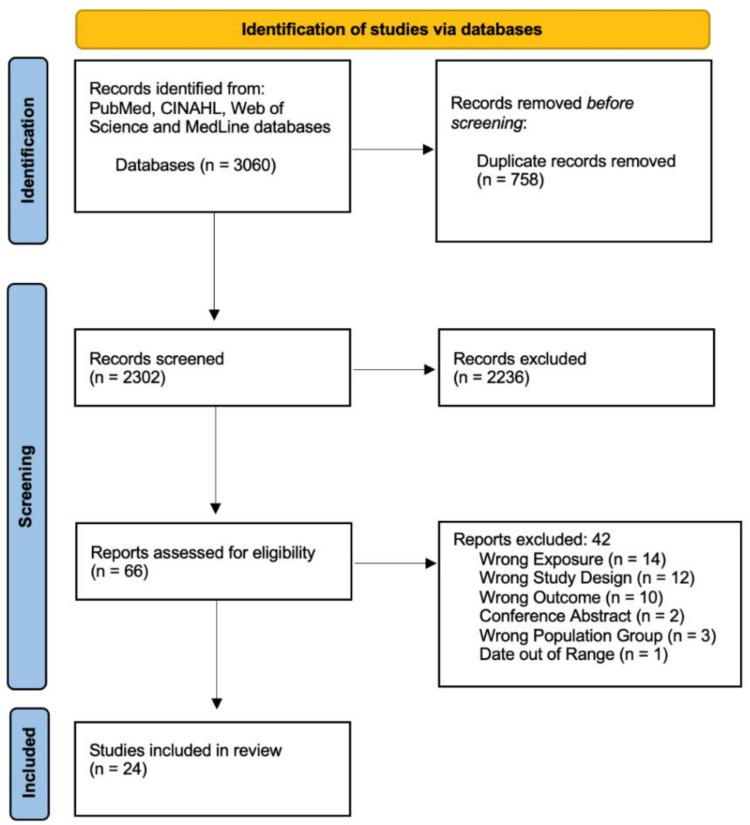
PRISMA flow diagram of included studies.

**Table 1 nutrients-13-03449-t001:** Database Search Strategy in CINAHL.

Search Strategy		
Number	Term/s	Limits
1	Medical Heading (MH): “child*”	
2	“child *”	Title and Abstract
3	MH: “obesity” OR “pediatric obesity” OR “body mass index” OR “waist circumference”	
4	“obesity” OR “obese” OR “overweight” OR “risk of overweight” OR “risk of obesity” OR “adiposity” OR “adipose” OR “excess body weight” OR “body mass index” OR “bmi” OR “central obesity” OR “waist circumference”	Title and Abstract
5	“infant growth” OR “rapid growth” OR “catch-up growth” OR “retarded growth” OR “growth velocity” OR “growth rate” OR “height velocity” OR “weight velocity” OR “weight-for-age” OR “height-for-age” OR “weight-for-length” OR “bmi-for-age”	Title and Abstract
6	1 OR 2	
7	3 OR 4	
8	5 AND 6 AND 7	2010—current, English, Subject age: child (6–12), preschool (2–5)

* Truncation, allows for multiple endings of a word to be searched.

**Table 2 nutrients-13-03449-t002:** Characteristics and data synthesis of included studies (n = 24).

		Sample		Infant Growth		Childhood Outcome			Size of Effect		
Study	Location	*N*	Weight Status	Age (months)	Growth Measure	Age (years)	Growth Measure	Analysis	Primary	Secondary	Risk of Bias (ADA Quality Checklist)
Jones-Smith et al. (2013). Prospective cohort study	Mexico	586	-	0–24	BMI velocity, length velocity, weight velocity	8	BAZ	Logistic regression	Overweight and infant BMI velocity, OR (95% CI): 1.47 (1.21, 1.78) Overweight and infant length velocity, OR (95% CI): 1.49 (1.23, 1.79) Overweight and infant weight velocity, OR (95% CI): 1.69 (1.40, 2.04) All values *p* < 0.05	-	Positive - High withdrawal rate
Lei et al. (2015). Prospective cohort study	USA	1957	SGA	0–12	Weight velocity, length velocity	7	Height, weight, BMI	Logistic regression	Overweight/obesity and excessive catch-up growth (SGA), OR (95% CI): 7.5 (5.4, 10.5)	-	Neutral - Generalisability - High withdrawal rate - Prediction calculation errors
Liu et al. (2017). Prospective cohort study	USA	1169	-	0–12	Weight, height, BMI trajectory	6	Weight, height, BMI	Logistic regression	Obesity and infant BMI trajectory, OR (95% CI): 1.82 (1.14, 2.89)	-	Neutral - Generalisability - Incomplete data sets - High withdrawal rate
Nguyen et al. (2021). Prospective cohort study	Vietnam	1402	-	0–12	Weight, length	6–7	Weight, height, HAZ, BMIZ	Multivariable linear and Poisson regression models	Overweight/obesity and infant weight gain, IRR (95% CI): 0–<6 mo: 1.35 (1.06, 1.72), *p* < 0.05 6–<12 mo: 1.40 (1.09, 1.81), *p* < 0.01	-	Positive - Low prevalence of overweight/ obesity among sample.
Peneau et al. (2011). Retrospective cohort study	France	998	Preterm infants were included in the sample.	0–24	Weight, length, BMI	7–9	Weight, height, BMI	Logistic regression model	Overweight and infant average monthly weight gain, OR (95% CI): Boys: 2.47 (1.32, 4.60), *p*: 0.01 Girls: 2.49 (1.38, 4.51), *p*: 0.001 Overweight and infant average monthly length gain, OR (95% CI): Girls: 2.15 (1.18, 3.92), *p*: 0.03	-	Positive - Inclusion of preterm children in the sample - Recall bias - Use of tertiles may have weakened the strength of association.
Penny et al. (2016). Longitudinal cohort study	Peru	1521	-	0–12	Weight, length	8	Weight, height, WC	Generalised estimating equations	Overweight/obesity and infant rapid weight gain, OR: Urban: 2.06, *p* < 0.001 Rural: 2.60, *p*: 0.001	-	Neutral - Generalisability - Inaccurate weight measurements
Shi et al. (2018). Retrospective cohort study	China	3004	SGA	0–24	Weight, length, WAZ, HAZ, WHZ	2–5	Weight, height, BMIZ, change in WAZ, HAZ, WHZ	Mixed-effects regression model	Overweight/obesity and infant excessive rapid catch-up growth, OR (95% CI): 11.6 (8.8, 15.3) Overweight/obesity and rapid catch-up growth, OR (95% CI): 2.3 (1.8, 3.0) *p* < 0.001	-	Positive - Confounding factors: feeding status, genetic potential, other health outcomes
Taveras et al. (2011). Longitudinal study	USA	44,622	-	0–24	Weight, length	5–10	Weight, height, BMI	Generalised linear mixed models	Obesity and infant growth trajectory (crossed upwards ≥2 weight-for-length percentiles), OR (95% CI): 5 years: 2.08 (1.84, 2.34) 10 years: 1.75 (1.53, 2.00)	-	Neutral - Incomplete data sets - Inaccurate anthropometric measurements - Generalisability
Nanri et al.(2016). Retrospective cohort study	Japan	1296		0–18	Weight, length	9–10	Weight, height, BMI	Logistic regression analysis	Overweight and rapid weight gain, adjusted OR (95% CI) Boys: 1.67 (0.83, 3.33) Girls: 2.60 (0.96, 7.04)		Neutral - Conflict of interest - Generalisability
Chirwa et al. (2014). Prospective cohort study	South Africa & Malawi	530	-	0–12	Weight velocity (kg/month), height velocity (cm/month), BMIZ	9–11	BMI, BMIZ	Linear and logistic regression	Overweight and infant weight velocity, OR (95% CI): 0–3 mo: 4.80 (2.49, 9.26), 0–6 mo: 2.60 (1.77, 3.83), 0–12 mo: 2.46 (1.89, 3.61) BMI and infant weight velocity, B (SE): 0–3 mo: 12.3 (1.05), 0–6 mo: 14.8 (1.23), 0–12 mo: 17.4 (1.37) BMIZ and infant height velocity, B (SE): 0–3 mo: 0.81 (0.25), 0–6 mo: 1.86 (0.50), 0–12 mo: 1.50 (0.54), all values *p* < 0.05	-	Neutral Sources of potential bias: - Incomplete data sets - High withdrawal rate
Taal et al. (2013). Prospective cohort study	The Netherlands	3531	SGA,AGALGA Preterm infants were included in the sample.	0–24	Weight, length	2–4	Weight, height, BMI	We used linear regression analysis to assess the associations of being SGA or LGA for birth weight with growth realignment.	BMIZ and infant catch-up growth, B (95% CI): SGA: −0.23 (−0.39, −0.07), *p* < 0.001 AGA: 0.44 (0.37, 0.52), *p* < 0.001 Overweight/obesity and infant catch-up growth, OR (95% CI): AGA: 3.11 (2.37, 4.08), *p* < 0.001	-	Neutral Sources of potential bias: - Incomplete data sets - Generalisability
Kagura et al. (2012). Prospective cohort study	South Africa	140	-	0–12	WAZ, HAZ, BMIZ	12	BMIZ	Regression models	BMIZ and infant Δ weight, B (95% CI): 0.19 (0.03, 0.35), SE: 0.06, *p* ≤ 0.01	-	Neutral - Convenient sampling - Conflict of interest
Polk et al. (2015). Retrospective cohort study	USA	463	-	0–24	Weight, length, WLZ	3	Weight, height, BMI	Mixed-effects models and multivariate linear regression	BMIZ and infant Δ WLZ, B (95% CI): Non Latino: Boys: 1.768 (1.34, 2.23) Girls: 1.031 (0.55, 1.51) Latino: Boys: 1.782 (1.17, 2.39) Girls: 1.430 (1.01, 1.85)	-	Neutral - Convenient sampling - Inaccurate anthropometric measurements - Generalisability
Kramer et al. (2014). Longitudinal cohort study	Republic of Belarus	13,879	SGA, AGA, LGA	0–12	Δ WAZ	6.5, 11.5	Height, weight, BMI, %BF, FMI, WC	Logistic regression	BMI and infant catch-up growth (>0.67 SDS), difference (95% CI): SGA: −0.2 (−0.4, 0.1)	%BF and infant catch-up growth (>0.67 SDS), difference (95% CI): SGA: −0.2 (−0.9, 0.4)	Positive - Generalisability
Zhou et al. (2016). Prospective cohort study	China	579	-	0–18	Δ WAZ	7–9	BAZ, MUAC, %BF, FMI using BIA	Multilevel mixed analysis	BAZ and rapid infant weight gain, B (95% CI): 0.69 (−0.49, 0.89), *p* < 0.001 Overweight/obesity and rapid infant weight gain, OR (95% CI): 2.94 (1.17, 7.43), *p*: 0.022	FMI and rapid infant weight gain, B (95% CI): 0.58 (0.37, 0.80) *p*: < 0.001	Positive - High withdrawal rate
Gishti et al. (2014). Prospective cohort study	The Netherlands	6464	-	0–12	Length gain, weight gain, abdominal circumference gain, BMI gain	6	BMI, FMI, LMI—using DXA and abdominal ultrasound	Linear regression models	BMI and infant growth rate, B (95% CI): Deceleration (D): −0.35 (−0.45, −0.26) Acceleration (A): 0.51 (0.41, 0.60) All values *p* < 0.01.	FMI and infant growth rate, B (95% CI): D: −0.22 (−0.32, −0.13) A: 0.25 (0.16, 0.35)	Positive - High withdrawal rate - Incomplete data sets
Golcalves et al. (2014). Prospective cohort study	Brazil	167	LBW ABW	0–6	Weight gain (SDS)	8	BMI, WC	Multivariate linear regression	BMI and infant weight gain (>0.67 SDS), B (95% CI): 1.4 (0.7, 2.2), *p*: 0.001	WC and infant weight gain (>0.67 SDS), B (95% CI): 4.0 (2.1, 5.9), *p* < 0.001	Positive - High withdrawal rate
Lin et al. (2021). Prospective cohort study	China	209	-	0–12	Δ WAZ	4	WAZ, weight, BMI, WC	Multivariate linear regression	BMI and infant rapid growth, B (95% CI): 0.78 (0.33, 1.23), *p* < 0.01	WC and infant rapid growth, B (95% CI): 1.62 (0.45, 1.32), *p* < 0.05	Neutral - Conflict of interest - Generalisability - High withdrawal rate - Recall bias
Nanri et al. (2017). Retrospective cohort study	Japan	439	-	0–18	Weight, length	9–10	Weight, height, %BF, WC	General linear model procedure	BMI and infant rapid weight gain, adjusted means Boys: 16.7 (16.2, 17.2) Girls: 16.5 (15.9, 17.0) BMI and infant weight gain (non-rapid) Boys: 16.1 (15.7, 16.6) Girls: 15.6 (15.1, 16.0) All values *p* < 0.05.	%BF and infant rapid weight gain, adjusted means Boys: 18.4 (17.3, 19.5) Girls: 16.3 (15.0, 17.5) WC and infant rapid weight gain, adjusted means Boys: 58.2 (56.8, 59.5) Girls: 56.6 (55.0, 58.1).	Neutral - Conflict of interest - Generalisability
Ong et al. (2020). Prospective cohort study	United Kingdom	254	-	0–24	Weight, length, skinfold thickness, WAZ	5–11	Weight, height, FM—DXA scan	Multilevel linear regression	BMI and infant Δ WAZ, B (SE): 0–3 mo: 0.28 (0.09), *p*: 0.002 3–24 mo: 0.26 (0.09), *p*: 0.001	%BF and infant Δ WAZ, B (SE): 0–3 mo: 1.38 (0.69), *p*: 0.0483–24 mo: 1.71 (0.64), *p*: 0.008	Positive - Self-reporting - Low ability of predictive models/equations
Woo et al. (2018). Retrospective cohort study	USA	346	-	0–24	Weight, length	3–7	BMIZ, LMI, FMI—using DXA scan, %BF	Linear and logistic regression	Overweight/obesity and infant weight velocity, OR (95% CI): 3 years: 1.07 (1.05, 1.10), *p* < 0.001 7 years: 1.03 (1.01, 1.05), *p* < 0.01	FMI and infant weight velocity, significant parameter estimate (SE): 0.02 (0.003), *p* < 0.001	Positive - Generalisability
Ejlerskov et al. (2015). Prospective cohort study	Denmark	233	-	0–9	Δ WAZ	3	BMI FFM and FMI—using BIASkinfolds	Multiple linear regression	-	FMI and infant Δ WAZ, B (SE): 0–5 mo: 0.21 (0.05) 5–9 mo: 0.42 (0.06) All values *p* < 0.001	Neutral - Measurement error - Prediction calculation errors - High withdrawal rate
Smith-Brown et al. (2018). Prospective cohort study	Australia	36		0–12	Weight, length, change in WLZ	2–3	Weight, height, WC, BMI, BMIZ, WHR, FFM, FM (D2O dilution technique)	Multiple linear regression	-	FMMI z-score and infant growth velocity, r: 0–6 mo: 0.57, *p*: 0.019 6–12 mo: −0.75, *p*: 0.019 FMI z-score and infant growth velocity, r: 6–12 mo: 0.75, *p*: 0.019	Neutral - Small sample size - Recall bias
Vogelezang et al. (2019). Prospective cohort study	The Netherlands	3205	-	0–12	Weight, length	10	Weight, height, BMI, FFM, FM—using DXA scan	Conditional and linear regression	-	Visceral fat and infant weight velocity, regression coefficient (95% CI): AGA: 0.04 (0.02, 0.06)	Neutral - Conflict of interest - Selection bias - Incomplete data sets

## Data Availability

No new data were created or analysed in this study. Data sharing is not applicable to this article.

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
