# Peer review of "The Effect of Growth Rate during Infancy on the Risk of Developing Obesity in Childhood: A Systematic Literature Review"

_nutrients, 2021, doi:10.3390/nu13103449_

Round 1
Reviewer 1 Report
Specific suggestions:
- Line 61: change the order between cm/month and g/month because in the sentence appear first, weight.
- Table 2: is it possible any other structure to help the comprehension?
- References:
- Skip the doble dot after the authors' name (1-12; 24).
- 10: There is not volume neither pages.
General doubt:
In the article by Rolland-Cachera et al. [12], it is said that the energy intake is one of the outcomes to investigate the risk of developing childhood obesity. Instead, this term do not appear in the discussion. Do you think that it is a factor to take into account in the studies and also what the mother and baby eat during pregnancy and breastfeeding?
Reviewer 2 Report
The systematic review by Halilagic and Moschonis deals with an important subject, namely the possible effect of infant growth on later risk of obesity. The review is well written, balanced and comprehensive.
Strengths
- The objectives, exposures and outcomes are clear and well described.
- The background in the introduction covers the main findings of previous research in this area.
- Both the exposure (infant growth) and outcome (measures of obesity) cover a narrow and well defined range which increases confidence in the study findings. For example, outcomes in children 2-12 years avoids the problems of confounding by puberty.
- The systematic review has been conducted to a high standard with clearly defined search strategy, and eligibility criteria.
- The results are well presented based on a logical order of outcomes (lines 230, 231).
Weaknesses
1.The review has few weaknesses. However, I am uncertain why the results and discussion are confined to those studies which report statistically significant findings. For example, the results by Nanri were not statistically significant and so were not included (lines 368 and 369). This is inherently a source of bias since statistical significance quite rightly is not a study eligibility criterion in section 2.1. To the best of my knowledge, statistical significance is not usually a criteria for a study results to be included in a systematic review. Consequently the authors may also want to focus more on the studies that did not show significance and discuss why this might be the case.
- For analyses of effects of secondary factors affecting the association between infant growth and later obesity (LINES 393 – 402) such as gender and the most sensitivity period (ie age) for the effects of infant growth were formal tests of statistical interaction conducted. For example, is the analysis ‘ the greater positive association between in infant change in WLZ and childhood BMIZ among boys than girls’ based on studies using interaction models.
